

# Interaction quenches in Bose gases studied with a time-dependent hypernetted-chain Euler-Lagrange method

**Mathias Gartner[1], David Miesbauer[1], Michael Kobler[2],**
**Julia Freund[1,3], Giuseppe Carleo[4] and Robert E. Zillich[1⋆]**

**1** Institute for Theoretical Physics, Johannes Kepler University Linz,
Altenberger Straße 69, 4040 Linz, Austria
**2** Institute for Quantum Gravity, Theoretical Physics III, Department of Physics,
Friedrich-Alexander-Universität Erlangen-Nürnberg,
Staudtstraße 7, 91058 Erlangen, Germany
**3** Institute of Semiconductor and Solid State Physics, Johannes Kepler University Linz,
Altenberger Straße 69, 4040 Linz, Austria
**4** Institute of Physics, École Polytechnique Fédérale de Lausanne (EPFL),
CH-1015 Lausanne, Switzerland

⋆ robert.zillich@jku.at

## Abstract

We present a new variational method to study the dynamics of a closed bosonic many-body system, the time-dependent hypernetted-chain Euler-Lagrange method, tHNC. Based on the Jastrow ansatz, it accounts for quantum fluctuations in a non-perturbative way. The tHNC method scales well with the number of dimensions, as demonstrated by our results on one-, two-, and three-dimensional systems. We apply the tHNC method to interaction quenches, i.e. sudden changes of the interaction strength, in homogeneous Bose gases. When the quench is strong enough that the final state has roton excitations (as found and predicted for dipolar and Rydberg-dressed Bose-Einstein condensates, respectively), the pair distribution function exhibits stable oscillations. For validation, we compare tHNC results with time-dependent variational Monte Carlo results in one and two dimensions.

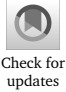

# 1   Introduction

The dynamics of many-body systems far from equilibrium and the role of interactions is an interesting and intensely studied topic. Many phenomena not known from linear response dynamics have been predicted and/or observed: many-body localization, where interactions may prevent self-equilibration if a system with disorder starts far from equilibrium [1], dynamical phase transitions [2] characterized by a non-analytical time evolution after a quench near a quantum phase transition point [3, 4], orthogonality catastrophe of polarons after an interaction quench [5], and non-thermal fixed points predicted in relaxation dynamics [6, 7]. In particular, the high level of control in experiments with ultracold quantum gases, either as continuous gases in harmonic [8] and box traps [9] or as lattice gases [10], facilitates the study of far from equilibrium dynamics [11, 12], by e.g. quenching an optical lattice [13] to study correlation dynamics, or by interaction quenches of Bose gases by Feshbach resonances [14] to study three-body correlations [15] and universality [16]. In the latter cases, a rapid quench to large $s$-wave scattering length $a$ is essential to investigate strongly interacting Bose-Einstein condensates (BEC), because equilibrium studies are hampered by losses due to three-body collisions when $\rho a^3$ becomes large, where $\rho$ is the number density.

Theoretical studies of many-body dynamics far from equilibrium have been performed by a variety of methods, some of which are best suited for lattice systems, like time-evolving block decimation (TEBD) [17], the application of the time-dependent variational principle (TDVP) [18] to matrix product states (MPS) [19], non-equilibrium dynamical mean-field theory (DMFT) [20], and the time-dependent density matrix renormalization group method (tDMRG) [21–23]. Some of them work best in one dimension, like methods based on continuous matrix product states (cMPS) [24], while others scale well to two and three dimensions, such as multiconfigurational time-dependent Hartree approaches (MCTDHF) [25].

The hypernetted-chain Euler-Lagrange (HNC-EL) method has been formulated for finding optimized ground states [26–28] and the dynamics in the linear response regime, the latter also termed correlated basis function method [29]. In this work, we derive an efficient time-dependent variational method for continuous Bose systems in any dimension by generalizing the HNC-EL method to a fully time-dependent method. This time-dependent hypernetted-chain Euler-Lagrange (tHNC) method is based on a Jastrow ansatz for the wave function like the ground state HNC-EL method, and akin to the time-dependent variational Monte Carlo (tVMC) method [30–32]. The tHNC method, however, can be orders of magnitude more efficient computationally because it does not require Monte Carlo sampling.

In this work, we use tHNC to study the dynamics of a homogeneous Bose gas after a sudden interaction quench. We are interested in the short period of time after the quench where three-body losses are not dominant yet, therefore we can neglect these losses and have a closed quantum system. Our primary interest is the time evolution of the pair distribution function $g(\mathbf{r}, t)$ after the interaction quench, in particular after a quench to a strongly correlated sys-

tem exhibiting roton excitations, which have zero group velocity. To assess the validity of the approximations of tHNC we compare our results in one and two dimensions with tVMC simulations.

## 2 The time-dependent hypernetted-chain Euler-Lagrange method

We consider a Bose gas of particles with positions $\mathbf{r}_i$ and mass $m$ in $d$ dimensions, interacting via a pair potential $v$. In equilibrium, such a system is described by a time-independent Hamiltonian

$$H_0 = -\frac{\hbar^2}{2m} \sum_{j=1}^{N} \Delta_j + \frac{1}{2} \sum_{k \neq l}^{N} v(\mathbf{r}_k - \mathbf{r}_l). \tag{1}$$

In this work, we consider completely homogeneous systems, which for quantum gases can be approximately realized by box traps [9]. We do not model the box trap boundaries where the density falls off rapidly; instead, we consider only the constant-density part in the interior of the trap. Consequently, the Hamiltonian $H_0$ for our model system does not contain an external one-body potential.

For Bose symmetry, the ground state wave function $\Phi_0(\mathbf{r}_1, \ldots, \mathbf{r}_N)$ can be readily calculated using, for example, exact quantum Monte Carlo simulations. Variational approximations to $\Phi_0$ can be obtained with less computational effort, such as variational Monte Carlo, including recent advances using artificial neural networks [33].

A well-established and straightforward variational treatment that includes correlations, i.e. "quantum fluctuations", in a non-perturbative way, are based on the Jastrow-Feenberg ansatz and its generalizations. The many-body Bose ground state $\Phi_0$ can be expressed in terms of two-body, three-body, etc. correlations,

$$\Phi_0(\mathbf{r}_1, \ldots, \mathbf{r}_N) = \frac{1}{\sqrt{\mathcal{N}}} \exp\Big[ \frac{1}{2} \sum_{k<l} u_2^{(0)}(\mathbf{r}_k - \mathbf{r}_l) + \frac{1}{3!} \sum_{k<l<m} u_3^{(0)}(\mathbf{r}_k, \mathbf{r}_l, \mathbf{r}_m) + \ldots \Big],$$

where $\mathcal{N}$ denotes the normalization integral $\langle \Phi_0 | \Phi_0 \rangle$. The real-valued correlation functions $u_n^{(0)}$ are obtained from the Ritz variational principle, which requires that the energy expectation value $E = \langle \Phi_0 | H | \Phi_0 \rangle / \langle \Phi_0 | \Phi_0 \rangle$ is minimized. The series of correlations has to be truncated for practical calculations. The Euler-Lagrange equations resulting from functional optimization, $\frac{\delta E}{\delta u_n^{(0)}} = 0$, involve high-dimensional integrals, which can be evaluated approximately using diagrammatic methods [34].

These equations are the hypernetted-chain Euler-Lagrange (HNC-EL) equations (the name becomes clear below). If two-body and three-body correlations are considered, the ground state energy and structural properties of strongly correlated systems, such as liquid $^4$He, are very close to exact Monte Carlo results [26]. For a less strongly correlated system, two-body correlations are sufficient.

Excitations of the many-particle system described by $H_0$ can be obtained from linear response theory, by allowing for small time-dependent fluctuations of the correlations, $u_n(\mathbf{r}_k, \mathbf{r}_l, \ldots, t) = u_n^{(0)}(\mathbf{r}_k, \mathbf{r}_l, \ldots) + \delta u_n(\mathbf{r}_k, \mathbf{r}_l, \ldots, t)$, and expanding the Euler-Lagrange equations up to linear order in $\delta u_n$, see Ref. [35] for details. Typically, excitations are generated by probing the system with a weak external time-dependent one-body potential, $\sum_j v_{\text{ext}}(\mathbf{r}_j, t)$ (such as a laser, neutrons etc.). Then one-body "correlations" $\delta u_1(\mathbf{r}_j, t)$ need to be included as well. Excellent agreement with experiments can be achieved, for example, for the dispersion relations of collective excitations in $^4$He if fluctuations of three-body correlations are taken into account [29].

An interesting question is what happens if the external perturbation is not weak, such that $\delta u_n(\mathbf{r}_k, \mathbf{r}_l, \ldots, t)$ cannot be assumed to be a small fluctuation around the ground state $u_n^{(0)}(\mathbf{r}_k, \mathbf{r}_l, \ldots)$ anymore. There are many examples of nonlinear response, such as nonadiabatic alignment of molecules [36,37], dynamic material design [38,39], or rapid parametric changes in ultracold gases such as interaction quenches [15,40], the latter being the focus of the present work. The nonlinear response of the system could be captured by expanding the Euler-Lagrange equations to higher orders in $\delta u_n$. Instead of following this path, we want to formulate the Euler-Lagrange equations for general time-dependent, complex correlations $u_n(\mathbf{r}_k, \mathbf{r}_l, \ldots, t)$, in order to find the time-dependent many-body wave function $\Phi(\mathbf{r}_1, \ldots, \mathbf{r}_N, t)$ as an approximate solution of the time-dependent Schrödinger equation $H(t)\Phi = i\hbar\dot{\Phi}$. In this work, we focus on perturbations caused by changing the interaction potential, which is modeled with the time-dependent Hamiltonian

$$H(t) = -\frac{\hbar^2}{2m} \sum_{j=1}^{N} \Delta_j + \frac{1}{2} \sum_{k \neq l}^{N} v(\mathbf{r}_k - \mathbf{r}_l, t). \tag{2}$$

No assumption about the magnitude of variations of $v$ in time will be made. Since we focus here on time-dependent interactions instead of time-dependent external perturbation potentials, we restrict ourselves to homogeneous systems. The interaction is translationally invariant, therefore the system remains homogeneous despite the variation of $v$. Regarding experimental realizations, Feshbach resonances are one of the means to vary the effective interaction over many orders of magnitude in experiments with ultracold quantum gases.

As long as the system is not too strongly correlated, two-body correlations are usually sufficiently accurate. Therefore, we restrict ourselves to two-body correlations $u_2(\mathbf{r}_k - \mathbf{r}_l, t)$ to discuss the dynamics resulting from a quench of $v(\mathbf{r}_k - \mathbf{r}_l, t)$. It turns out to be convenient to split $u_2$ into its real and imaginary part, $u_2(r) \equiv u(r) + 2i\varphi(r)$, where $r = |\mathbf{r}_k - \mathbf{r}_l|$ is the distance between particle $k$ and $l$. The time-dependent generalization of the Bose Jastrow-Feenberg ansatz is

$$\Phi(\mathbf{r}_1, \ldots, \mathbf{r}_N, t) = \frac{1}{\sqrt{\mathcal{N}(t)}} \exp\left[\frac{1}{2} \sum_{k<l} u(|\mathbf{r}_k - \mathbf{r}_l|, t) + i \sum_{k<l} \varphi(|\mathbf{r}_k - \mathbf{r}_l|, t)\right]. \tag{3}$$

Since $u$ and $\varphi$ depend on time, all quantities introduced below depend on time as well.

The Euler-Lagrange equations of motion for $u$ and $\varphi$ are obtained from the generalization of the Ritz variational principle to the time-dependent Schrödinger equation, the minimization of the action integral

$$\mathcal{S} = \int_{t_0}^{t} dt' \mathcal{L}(t'), \tag{4}$$

with the Lagrangian

$$\mathcal{L}(t) = \langle \Phi(t) | H(t) - i\hbar \frac{\partial}{\partial t} | \Phi(t) \rangle. \tag{5}$$

The second expression in $\mathcal{S}$ involving the time derivative can be simplified by the invariance of $\mathcal{S}$ with respect to adding total time derivatives to $\mathcal{L}$. Terms involving the kinetic energy operator can be simplified with the Jackson-Feenberg identity for a real-valued $F$

$$F\Delta F = \frac{1}{2}\left(\Delta F^2 + F^2 \Delta\right) + \frac{1}{2} F^2 [\nabla, [\nabla, \ln F]] - \frac{1}{4}[\nabla, [\nabla, F^2]],$$

where $F = e^{\frac{1}{2}\sum_{k<l} u(|\mathbf{r}_k - \mathbf{r}_l|)}$ in our case. The Lagrangian $\mathcal{L}$ can be brought into a convenient form,

$$\mathcal{L} = \mathcal{L}_g + \mathcal{L}_3 + \frac{\hbar}{2} \int d^d r \, \dot{\varphi}(r, t) g(r, t) + \frac{\hbar^2}{2m} \rho \int d^d r \, \mathbf{v}(\mathbf{r}, t)^2 g(r, t), \tag{6}$$

with

$$\mathcal{L}_g = \frac{\rho}{2} \int d^d r \Big[ v(r,t) - \frac{\hbar^2}{4m} \Delta u(r,t) \Big] g(r,t),$$

$$\mathcal{L}_3 = \frac{\hbar^2}{2m} \rho^2 \int d^d r \, d^d r' \, \mathbf{v}(\mathbf{r},t) \cdot \mathbf{v}(\mathbf{r}',t) \, g_3(r,r',|\mathbf{r}-\mathbf{r}'|,t).$$

We abbreviated the gradient of the phase as $\mathbf{v}(\mathbf{r},t) \equiv \nabla \varphi(r,t)$. The pair and three-body distribution functions, $g(r,t)$ and $g_3(r,r',|\mathbf{r}-\mathbf{r}'|,t)$, are expressed as function of distance vectors (where $\mathbf{r} = \mathbf{r}_1 - \mathbf{r}_2$ and $\mathbf{r}' = \mathbf{r}_1 - \mathbf{r}_3$ are distances between the particles at $\mathbf{r}_1$, $\mathbf{r}_2$, and $\mathbf{r}_3$). They are obtained from the corresponding pair and three-body densities via

$$\rho_2(r,t) = \rho^2 g(r,t),$$

$$\rho_3(r,r',|\mathbf{r}-\mathbf{r}'|,t) = \rho^3 g_3(r,r',|\mathbf{r}-\mathbf{r}'|,t),$$

where the $n$-body density $\rho_n$ is defined as

$$\rho_n(\mathbf{r}_1,\ldots,\mathbf{r}_n,t) = \frac{N!}{(N-n)!} \int dr_{n+1} \ldots dr_N |\Phi(t)|^2.$$

For a homogeneous system, $\rho_1 \equiv \rho$ is the (constant) number density.

We vary $\mathcal{S}$ by functional derivation, for which we need the relation between $u$ on the one hand and $g$ and $g_3$ on the other hand. Since $g$ (and $g_3$) does not depend on $\varphi$, $g$ and $u$ are related via the hypernetted-chain equation [34] just as in ground state HNC-EL calculations,

$$g(r,t) = e^{u(r,t)+N(r,t)+E(r,t)}, \tag{7}$$

where $N(r,t)$ are the so-called nodal diagrams and $E(r,t)$ the elementary diagrams. The former can in turn be expressed in terms of $g$ via the Ornstein-Zernicke relation, while the latter have to be approximated by truncating the infinite series of elementary diagrams. Details on these diagrammatic summations can be found in Ref. [26, 34]. Note that the contribution $\mathcal{L}_g$, which does not depend on $\varphi$, is just the expression for the energy expectation value also used in the ground state optimization of the HNC-EL method.

Unlike the ground state energy expectation value for the Jastrow-Feenberg ansatz with pair correlations, the Lagrangian $\mathcal{L}$ for the time-dependent problem contains the three-body distribution $g_3$, which is a functional of $u$ and hence of $g$, but cannot be given in a closed form. Two approximations of $g_3$ are common [41]: the convolution approximation which reproduces the correct long-range behavior of $g_3$, and the Kirkwood superposition approximation

$$g_3(r,r',|\mathbf{r}-\mathbf{r}'|,t) \approx g(r,t)\, g(r',t)\, g(|\mathbf{r}-\mathbf{r}'|,t), \tag{8}$$

which reproduces the correct short-range behavior. Both approximations can be systematically improved. Since we are interested here in impulsive changes of the short-range repulsion of the interaction, we choose the latter approximation.

The time-evolution of the time-dependent Jastrow-Feenberg ansatz $\Phi(t)$, eq. (3), is determined by solving the time-dependent hypernetted-chain Euler-Lagrange (tHNC) equations

$$\frac{\delta \mathcal{S}}{\delta g(r,t)} = 0, \tag{9}$$

$$\frac{\delta \mathcal{S}}{\delta \varphi(r,t)} = 0. \tag{10}$$

The tHNC equations are nonlinear partial differential equations. They could be cast into a form which resembles the Navier-Stokes equations, similar to the Madelung formulation of

the one-body Schrödinger equation, but we opt instead for a formulation in terms of a suitably defined "wave function" that is numerically more convenient.

For numerical solution of eqns. (9) and (10), we define an effective "pair wave function"

$$\psi(r,t) \equiv \sqrt{g(r,t)} e^{i\varphi(r,t)}.$$

Within the Kirkwood superposition approximation we can then cast the two real-valued eqns. (9) and (10) into a single complex nonlinear Schrödinger-like equation for $\psi(r,t)$. The derivation can be found in the appendix. The final form of the tHNC equation is

$$i\hbar \frac{\partial}{\partial t} \psi(r,t) = -e^{-i\gamma(r,t)} \frac{\hbar^2}{m} \nabla^2 e^{i\gamma(r,t)} \psi(r,t) + [v(r,t) + w_I(r,t)] \psi(r,t)$$
$$+ \left[ \beta(r,t) - \frac{\hbar^2}{m} |\nabla\gamma(r,t)|^2 \right] \psi(r,t). \tag{11}$$

The induced interaction $w_I$ is a functional of $g(r)$ only, and it is the same expression that appears also in ground state HNC-EL calculations, see e.g. Ref. [28]. This induced interaction can be interpreted as a phonon-mediated interaction in addition to the bare interaction $v$. An additional potential term appears if the elementary diagrams $E$ mentioned above are taken into account; since $E$ depends only on $g$, the same approximations for $E$ as in the ground state HNC-EL method could be applied. For simplicity we neglect $E$ in this work. The expressions $\beta$ and $\gamma$ are functionals of both $g(r)$ and $\varphi$,

$$\beta(r,t) = \frac{\hbar^2}{m} \rho \int d^d r' \, \mathbf{f}(\mathbf{r}',t) \cdot \mathbf{f}(\mathbf{r}'-\mathbf{r},t), \tag{12}$$

$$\nabla\gamma(r,t) = \rho \int d^d r' \, \mathbf{f}(\mathbf{r}',t) [g(\mathbf{r}'-\mathbf{r},t) - 1], \tag{13}$$

with $\mathbf{f}(\mathbf{r},t) \equiv g(r,t)\mathbf{v}(\mathbf{r},t)$. Note that $\gamma$ in eq. (11) is calculated by integrating $\nabla\gamma(r,t)$ given in eq. (13).

The formulation (11) is chosen because it can be solved with standard techniques such as operator splitting methods [42]. In the zero-density limit $\rho \to 0$, all many-body effects vanish: $\gamma \to 0$, $w_I \to 0$, $\beta \to 0$, hence eq. (11) becomes the bare two-body scattering equation for a two-body wave function $\psi$ with reduced mass $\frac{m}{2}$. We remind that, in this paper, we have an isotropic, homogeneous interaction $v(|\mathbf{r}_1 - \mathbf{r}_2|, t)$ and we assume that the system remains isotropic and homogeneous for all times, i.e. it never spontaneously breaks translation symmetry. Lifting the restriction of homogeneity and isotropy is formally straightforward, but solving the resulting tHNC equations would be computationally much more demanding.

## 3   Results

We present results for a homogeneous $d$-dimensional gas of bosons, where $d = 1, 2,$ and 3. The interactions $v(r,t)$ are either simple models for a repulsive interaction or interactions between Rydberg-dressed atoms [43]. In all cases, $v(r,t)$ is characterized by two parameters: an interaction range $R$ and an interaction strength $U$, see below. The system is in the ground state $\Psi_0$ for times $t < 0$, with interaction parameters $R_0$ and $U_0$. At $t = 0$, we switch either the width parameter $R$ to a new value, $R(t) = R_0 + (R_1 - R_0)\Theta(t)$, or make a similar switch of $U$. For $t > 0$, the previous ground state $\Psi_0$ evolves according to the new Hamiltonian $H$ characterized by the new interaction. At $t = 0$ the energy changes abruptly, but for $t > 0$ the evolution is unitary and thus energy is conserved, because $H$ is time-independent after the quench. Since there is no external potential and the interaction is translationally invariant, the

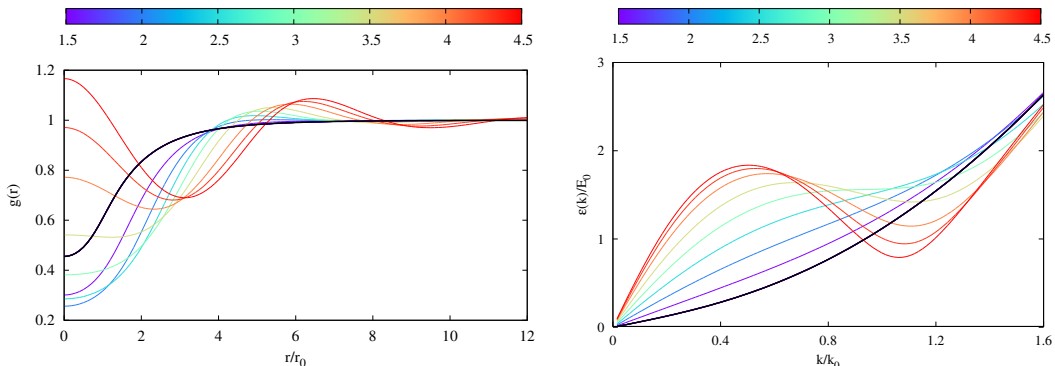

Figure 1: Left: pair distribution function $g(r)$ for the ground state of a three-dimensional Rydberg-dressed Bose gas for several range parameters $R$ and for fixed interaction strength $U/E_0 = 2$, see eq. (14). The thick black curve is for $R/r_0 = 1$, which is the initial value from which we quench to larger $R_1$ in the dynamical calculations. Other curves are for $R/r_0 = 1.5; 2.0; 2.5; 3.0; 3.5; 4.0; 4.3; 4.5$ (the colors correspond to different $R$, given by the colorscale), which correspond to the target values $R_1$ to which we quench in the dynamical calculations. Right: the Bijl-Feynman excitation spectrum $\varepsilon_F(k)$ for the above values of $R$.

system is homogeneous before the quench; precluding symmetry breaking, the system stays homogeneous after the quench.

In section 3.1 we present tHNC results of the pair distribution function $g(r, t)$ after a quench in a Rydberg-dressed Bose gas in three dimensions, where we show that roton excitations play an important role for the time-evolution of the pair distribution function. Rydberg-dressed Bose gases have been studied theoretically quite extensively [44–46], including studies of the dynamics after interaction quenches in the Bogoliubov approximation [47], and calculations of the roton excitation spectrum [45, 46]. The influence of roton excitations on the dynamics after interaction quenches have been studied for dipolar gases [48], again in Bogoliubov approximation. In appendix C, we compare the ground state $g(r)$ for the 3D Rydberg gas obtained within the Bogoliubov approximation to results obtained with HNC-EL and with exact path integral Monte Carlo (PIMC) simulations in the low temperature limit from [46]. The comparison shows that HNC-EL agrees very well with the exact PIMC result, while the Bogoliubov approximation deviates quite strongly from the exact result.

In order to assess the approximations of the tHNC method, we compare the dynamics of $g(r, t)$ with tVMC results in two and one dimensions in section 3.2, using the same Jastrow ansatz as in tHNC, but with fewer approximations. In general, correlations play a larger role in lower dimensional systems, hence these comparisons are a harder test of tHNC than in three dimensions, and furthermore, 3D tVMC simulations would have been computationally even more expensive.

## 3.1 Quench dynamics in 3D

Rydberg-dressing means that the ground state and a Rydberg state of the atoms are coupled by a laser detuned from resonance. The coupled potential energy surfaces of the ground states and the Rydberg states can be described by the following pair interaction [43, 44, 49]

$$v(r, t) = \frac{U(t)}{1 + (r/R(t))^6},$$  (14)

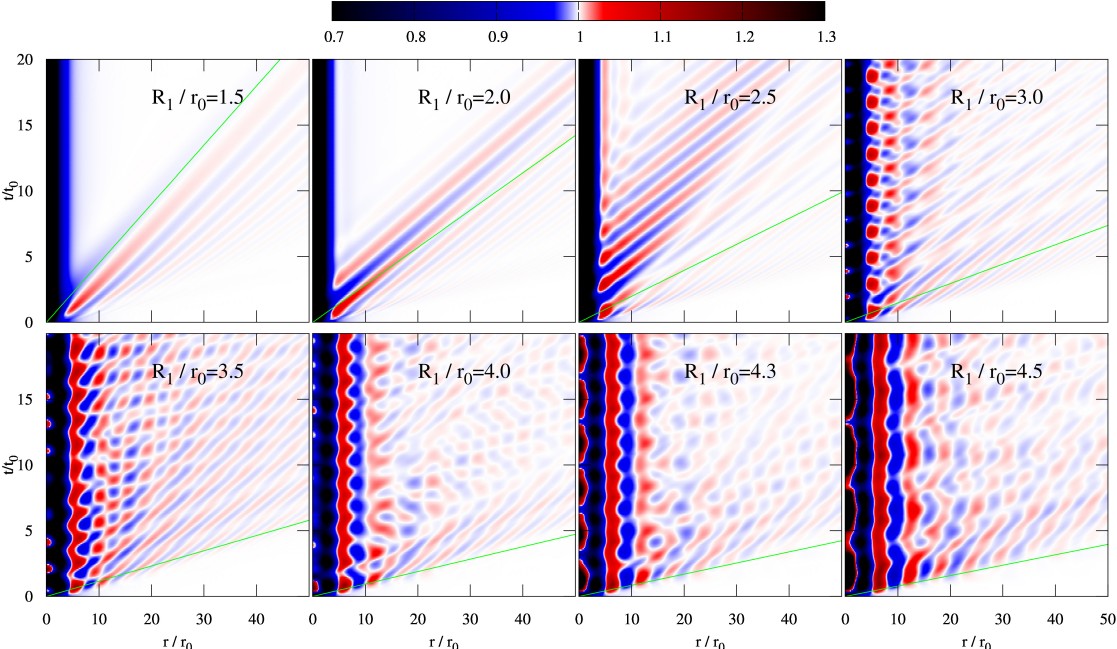

Figure 2: Pair distribution function $g(r,t)$ of a 3D Rydberg-dressed Bose gas after a quench from $R_0/r_0 = 1$ to $R_1/r_0 = 1.5; 2.0; 2.5; 3.0; 3.5; 4.0; 4.3; 4.5$, with fixed $U/E_0 = 2$. Blue indicates $g(r,t) < 1$ and red indicates $g(r,t) \geq 1$ (see color scale). For $R_1/r_0 \approx 3.0$ and higher, the oscillations of $g(r,t)$ near $r = 0$ do not decay anymore due to generation of roton pairs with vanishing group velocity. The green line shows the "sound cone" $r = 2ct$, where $c$ is the speed of sound after the quench.

where the strength $U(t)$ and the range $R(t)$ may depend on time. When two particles are closer than $\approx R$, they are only weakly repelled because $v$ becomes flat for small $r$; for $r \gtrsim R$, they feel a van der Waals repulsion. We follow Ref. [46] and measure wave numbers in units of $k_0 = (6\pi^2 \rho)^{1/3}$ in 3D, corresponding to a length unit $r_0 = 1/k_0$ (i.e. the density is always $\rho r_0^3 = (6\pi^2)^{-1}$). Energy is measured in units of $E_0 = \frac{\hbar^2 k_0^2}{2m}$, and time in units $t_0 = \frac{\hbar}{E_0}$.

We study quenches from weak to strong interactions. We keep $U$ fixed at $U_0 = 2E_0$, but switch $R(t)$ at $t = 0$, from $R_0/r_0 = 1$ to $R_1/r_0 = 1.5; 2.0; 2.5; 3.0; 3.5; 4.0; 4.3; 4.5$. In the left panel of Fig. 1 we show the corresponding *ground state* pair density distributions $g(r)$. The spatial oscillations in $g(r)$ become more pronounced as $R_1$ grows, and the range of correlations increases. Typically for the interaction (14), particles tend to cluster for larger $R$, eventually leading to a cluster solid [44]. This tendency to cluster is seen in the growth of $g(r)$ for small distance $r$ as we increase $R_1$.

The time evolution of $g(r,t)$ after a quench from $R_0/r_0 = 1$ to the target values is shown in Fig. 2 as color maps, where the horizontal axis is the distance $r$ and the vertical axis is the time $t$. The evolution of $g(r,t)$ shows that the information about the quench of the pair interaction is spreading to larger distances $r$ as time evolves. Lieb and Robinson proposed a "light cone" outside of which the effect of the quench has not yet arrived [50]. This light cone bound applies only to discrete Hamiltonians, and was found for the Bose Hubbard model in Ref. [31] with tVMC in one and two dimensions, where the "light" are the elementary excitations. Such a light cone appears if the group velocity $v_g$ of the elementary excitations has an upper bound $c$. Then a quench can excite two waves in opposite directions (conserving total momentum zero); the information about the quench would travel with $2c$, hence the response of $g(r,t)$ is expected to move with $2c$ to larger $r$.

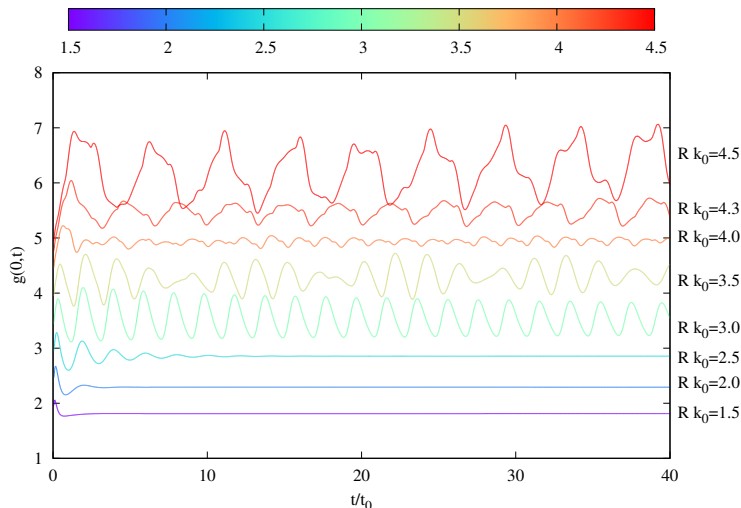

Figure 3: The pair distribution at $r = 0$, $g(0, t)$, after a quench from $R_0/r_0 = 1$ to $R_1/r_0 = 1.5; 2.0; 2.5; 3.0; 3.5; 4.0; 4.3; 4.5$, with fixed $U/E_0 = 2$. For better visibility, the curves are shifted with respect to each other and colored based on $R_1$, see colorscale at the top of the figure. For $R_1/r_0 = 3$ and higher, $g(0, t)$ oscillates with no apparent decay, while for smaller $R_1$, the pair distribution at zero distance equilibrates to a constant value.

The green lines in Fig. 2 show the light cone based on the speed of sound $c$ of the Rydberg-dressed Bose gas, the "sound cone" given by $r = 2ct$. Especially up to $R_1 = 3r_0$ the waves in $g(r, t)$ move faster than $2c$, and are not bound by the sound cone. This is not surprising since $c$ is not the highest group velocity. Especially for low $R_1$, this can be seen in the right panel of Fig. 1. There we show the excitation spectrum $\varepsilon_F(k)$ for the initial $R_0$ before the quench (thick line) and for the target values $R_1$ (colored lines), using the Bijl-Feynman approximation [51, 52], $\varepsilon_F(k) = \frac{\hbar^2 k^2}{2m S(k)}$. The Bijl-Feynman spectrum is calculated with the ground state static structure factor $S(k)$ obtained from a ground state HNC-EL/0 calculation with the respective value $R_1$. For larger $R_1$, the dispersion relation $\varepsilon_F(k)$ becomes steep, corresponding to a large $c$. In this regime, the evolutions of $g(r, t)$ approximately obey the sound cone bound, but upon closer inspection one can see small oscillations with large wave number which spread faster than $2ct$. Hence, the sound cone is not a strict bound. Again, this is not surprising because, at least in the Bijl-Feynman approximation, the dispersion relation of the Rydberg-dressed Bose gas has arbitrarily large group velocities for large $k$, because $\varepsilon_F(k) \to \frac{\hbar^2 k^2}{2m}$ for $k \to \infty$. Note that this is different from excitations in lattice Hamiltonians, characterized by quasi-momenta within the finite Brillouin zone, and thus $v_g$ does have a maximum (in the usual single-band Hubbard approximation). The group velocity $v_g$ is usually the sound velocity, which becomes the maximal speed of information spreading.

For small target values $R_1$, $g(r, t)$ quickly converges to an equilibrium distribution in an $r$-interval that grows with time, as the perturbation travels away to large $r$. The equilibrium is not the ground state $g(r)$ at the same $R_1$, because the quench injects energy into the system. For example for the quench from $R_0 = 1r_0$ to $R_1 = 2r_0$, the energy per particle jumps from $E = 0.073 E_0$ (the ground state energy before the quench) to $E = 0.739 E_0$ and then of course stays constant during the unitary time evolution; the *ground state* energy for $R_1 = 2r_0$, however, is lower, $E_g = 0.643 E_0$.

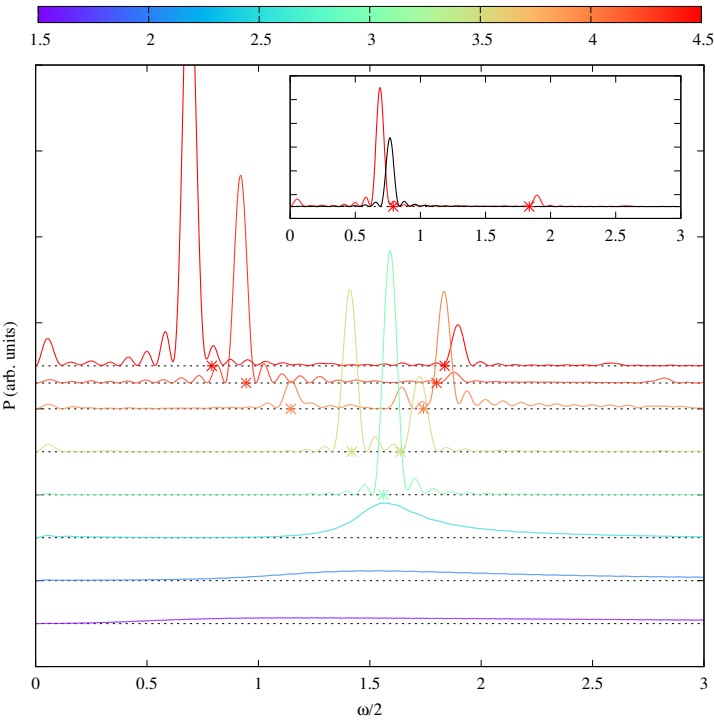

Figure 4: Power spectrum $P(\omega)$ of $g(0, t)$ shown in Fig. 3, after a quench from $R_0/r_0 = 1$ to $R_1/r_0 = 1.5; 2.0; 2.5; 3.0; 3.5; 4.0; 4.3; 4.5$, with fixed $U/E_0 = 2$. For better visibility, the curves are shifted with respect to each other and colored based on $R_1$, see colorscale at the top of the figure. The stars on the base line indicate the energies of the roton and the maxon after the quench, if present. The inset compares $P(\omega)$ after a quench to $R_1 = 4.5\, r_0$ starting from $R_0 = r_0$ (red) with a quench starting from $R_0 = 4\, r_0$ (black).

When we quench the interaction to larger $R_1$, there is a qualitative change in $g(r, t)$: we observe long-lived oscillations for $R_1 \geq 3\, r_0$ that do not decay within the time window shown in Fig. 2. This can be seen, for example, for small $r$. In Fig. 3 we show $g(r, t)$ for $r = 0$ for all target values $R_1$. The pair distribution function at $r = 0$ is one way to obtain the contact parameter [53, 54] which can be measured [55]. For $R_1 \leq 2.5\, r_0$, $g(0, t)$ converges to a constant value, which lies slightly above the ground state $g(0)$. For $R_1 \geq 3\, r_0$, $g(0, t)$ appears to keep oscillating indefinitely. Apart from $R_1 = 3\, r_0$, these oscillations clearly contain more than one frequency.

The origin of this long-lived oscillation for small $r$ becomes apparent when we invoke the picture of a quench that generates two opposite excitations. For small $R_1$, the Bijl-Feynman spectrum $\varepsilon_F(k)$ increases monotonically with wave number $k$, see right panel of Fig. 1. For $R_1 = 3\, r_0$, $\varepsilon_F(k)$ has a plateau with essentially zero slope around $k/k_0 = 1$, and for larger $R_1$, $\varepsilon_F(k)$ exhibits a maximum, called maxon, with energy $\hbar\omega_m$, and a minimum, called roton, with energy $\hbar\omega_r$. A vanishing group velocity $v_g(k) = \frac{d\varepsilon_F(k)}{dk}$ implies a diverging density of state, leading to a high probability to excite excitations with $v_g \approx 0$. Furthermore, the excitation pairs of opposite momenta produced by the quench do not propagate for $v_g = 0$. For $R_1 > 3\, r_0$, roton pairs as well as maxon pairs with opposite momenta are generated. Since they do not propagate, the temporal oscillations for small distance $r$ become long-lived for $R_1 \geq 3\, r_0$.

For $R_1 = 3\,r_0$, $g(0,t)$ oscillates with a single frequency because maxon and roton coincide at the inflection point of $\varepsilon_F(k)$. The frequency is twice the corresponding excitation energy (because the quench produces a *pair* of excitations). For $R_1 > 3\,r_0$, $g(0,t)$ oscillates with two frequencies, given by twice the roton and twice the maxon frequency. In order to confirm this quantitatively, we show the power spectra $P(\omega)$ of $g(0,t)$ in Fig. 4, as functions of $\omega/2$. Each $P(\omega)$ is shifted in proportion to $R_1$ for better visibility. For $R_1 < 3\,r_0$, the power spectra are broad. At $R_1 = 3\,r_0$ a single peak appears, corresponding to the single frequency oscillations seen in Fig. 3. For $R_1 > 3\,r_0$, $P(\omega)$ has two peaks of varying relative spectral weight: at twice the roton frequency $2\,\omega_r$ and at twice the maxon frequency $2\,\omega_m$ (the combination $\omega_r + \omega_m$ would have finite momentum and cannot be excited in this simple picture by a translationally invariant perturbation such as an interaction quench). The small ringing oscillations are artifacts from the Fourier transformation of a finite time window $[0, 40\,t_0]$.

However, $\omega_r$ and $\omega_m$, which are indicated by stars in Fig. 4 for the respective $R_1$, do not match perfectly with the peaks of $P(\omega)$. The picture of an interaction quench exciting two elementary excitations with opposite momenta is only approximately valid. A quench from $R_0/r_0 = 1$ to e.g. $R_1/r_0 = 4.5$ is a highly nonlinear process that cannot be regarded as a small perturbation and treated with linear response theory. The quench tends to shift the lower-frequency below $2\,\omega_r$ and the higher-frequency peak above $2\,\omega_m$. The effect of nonlinearity is demonstrated in the inset, where the power spectrum $P(\omega)$ for the quench $1\,r_0 \to 4.5\,r_0$ shown in the main figure (red) is compared with $P(\omega)$ for a much weaker quench $4\,r_0 \to 4.5\,r_0$ (black), where linear response theory may hold. From linear response theory, we would expect that, if two rotons are created, $g(r,t)$ will oscillate with exactly twice the roton frequency. This is indeed what we observe in the inset: the power spectrum has a peak very close to $2\,\omega_r$. Note that for the weaker quench, the excitation of two maxons is completely suppressed, because the quench injects less energy into the system. The difference between the energy/particle after the weak quench and the energy/particle of the $R_1/r_0 = 4.5$ ground state is just $\Delta E = 0.03\,E_0$, while an order of magnitude more energy is injected by the stronger quench, with an excess energy of $\Delta E = 0.54\,E_0$. More energy is available in the latter case to excite maxons, which have about twice the energy of rotons, see Fig.1.

## 3.2 Comparison with tVMC

We compare tHNC results for the quench dynamics with corresponding results obtained with time-dependent variational Monte Carlo (tVMC) simulations [30]. Details about our implementation of tVMC can be found in the appendix and in Refs. [56,57]. We use the same Jastrow ansatz (3) as in the tHNC method. However, tVMC does not require an approximation for $g_3$ nor does tVMC need to use approximations for the elementary diagrams, because all integrations over the $N$-body configuration space are performed by brute force Monte Carlo sampling. The price is, of course, a much higher computational cost. Therefore we have restricted the comparisons with tVMC to one and two dimensions.

### 3.2.1 Comparison in 1D

For the 1D comparison we use a square well interaction potential

$$v(r,t) = U(t)\Theta[R(t) - r]\,, \tag{15}$$

characterized by strength $U(t)$ and range $R(t)$. The length and energy units are $r_0 = \rho^{-1}$ and $E_0 = \frac{\hbar^2}{2mr_0^2}$. In the tVMC simulations we use $N = 100$ particles, corresponding to a simulation box size of $L = 100\,r_0$. We can thus calculate $g(r,t)$ up to a maximal distance of $r = 50\,r_0$; when fluctuations reach this maximal distance, spurious reflections appear due to the periodic

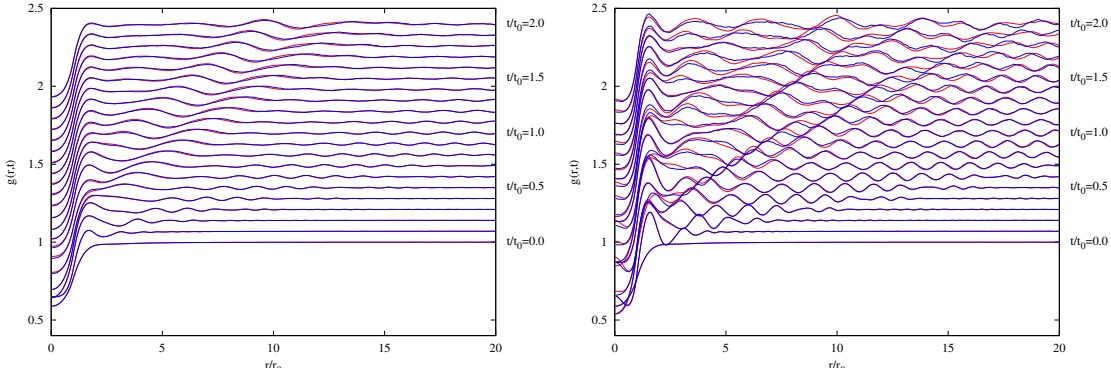

Figure 5: Comparison of the pair distribution function $g(r,t)$ for a Bose gas in 1D between tHNC (red) and tVMC (blue). The interaction is the square well potential (15), quenched from initial strength $U = 2E_0$ to a strength $U = 4E_0$ (left panel) and $U = 8E_0$ (right panel), respectively. For better visibility $g(r,t)$ is shifted in proportion to time, indicated on the right margin.

boundary conditions. Therefore, we restrict our comparisons of $g(r,t)$ with tHNC to times $t$ before these reflections become noticeable. Further technical details of the tVMC simulations can be found in the appendix.

In Fig. 5 we compare $g(r,t)$ after a quench. The interaction range is fixed at $R/r_0 = 1$ and the interaction strength jumps from $U = 2E_0$ to a target value $U = 4E_0$ (left panel) and to $U = 8E_0$ (right panel). We show $g(r,t)$ at times $t/t_0 = 0.0; 0.1; 0.2; \ldots; 2.0$. For the weaker first quench, the agreement between tHNC (red) and tVMC (blue) is excellent, because the target interaction is weak enough that neglecting elementary diagrams and the Kirkwood superposition approximation (8) for the three-body distribution $g_3$ are still good approximations. For the stronger quench to a target value $U = 8E_0$ the tHNC and tVMC results for $g(r,t)$ do not match perfectly anymore. For strong interaction, the effect of elementary diagrams or the three-body distribution or both becomes more important. But overall, the agreement is still remarkably good; for example, both frequency and phase shift of the oscillations in $g(r,t)$ are the same. We conclude that tHNC works quite well compared to tVMC in 1D, despite the approximations that we use in our simple implementation of tHNC. Of course, for strong interactions, even tVMC with pair correlations is not sufficient for quantitative predictions of the dynamics in 1D and a better variational ansatz than (3) should be used.

### 3.2.2 Comparison in 2D

For the 2D comparison we use the Rydberg-dressed interaction (14) from the 3D studies in the previous section 3.1. We use the 2D version of the units introduced above for 3D, see also [46]: wave numbers are in units of $k_0 = (4\pi\rho)^{1/2}$, and again length in units of $r_0 = 1/k_0$, energy in units of $E_0 = \frac{\hbar^2 k_0^2}{2m}$, and time in units of $t_0 = \frac{\hbar}{E_0}$. Again, we compare only up to times before effects of the periodic boundaries in tVMC contaminate the dynamics of $g(r,t)$.

Following the quench procedure in our 3D tHNC calculations above, we keep $U$ fixed at $U_0 = 2E_0$, and switch $R(t)$ from $R_0 = 2r_0$, where excitations from the ground state are monotonous, to $R_1 = 4r_0$, where the excitation spectrum exhibits rotons. Fig. 6 compares the tHNC result (red) and tVMC result (blue) for $g(r,t)$ after the quench. As in 1D, we find good agreement between tHNC and tVMC before spurious oscillations due to the periodic boundary conditions in tVMC appear for later times (not shown). As expected from our 3D calculations, the creation of pairs of rotons leads to persistent oscillations in $g(r,t)$ for small $r$. The ampli-

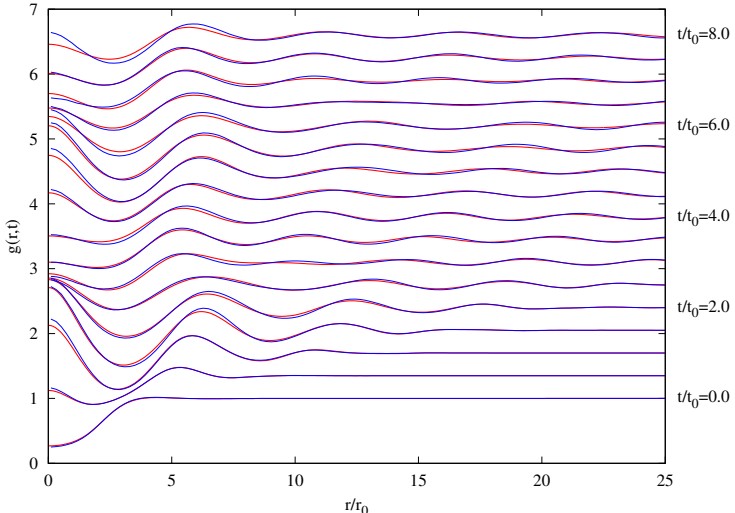

**Figure 6:** Comparison of the pair distribution function $g(r, t)$ for a Bose gas in 2D between tHNC (red) and tVMC (blue). The interaction is the Rydberg potential (14), quenched from initial range $R = 2\,r_0$ to $R = 4\,r_0$. For better visibility $g(r, t)$ is shifted in proportion to time, indicated on the right margin.

tude, however, becomes smaller in the tHNC results, as can be seen in Fig. 7, which compares $g(r, t)$ for $r = 0$. The deviation between the tHNC result and the tVMC result is larger than the stochastic error inherent in the tVMC method, which increases with time and is shown as shaded area in Fig. 7. Hence, the differences between tHNC and tVMC are due to the approximations made in our present implementation of tHNC. When $g(0, t)$ becomes large, particles tend to cluster together, and we expect that particularly the Kirkwood superposition form (8) for $g_3$ is a poor approximation.

# 4 Discussion

We present a new method for studying quantum many-body dynamics far from equilibrium, the time-dependent generalization of the hypernetted-chain Euler-Lagrange method, tHNC, which is non-perturbative and goes beyond the mean field paradigm. We demonstrate this variational method in a study of the interaction quench dynamics of a homogeneous Rydberg-dressed Bose gas. A sudden strong change of the effective Rydberg interaction leads to a strong response of the pair distribution function $g(r, t)$, which is the quantity we are interested in in this work. In an interaction quench, the Hamiltonian becomes time-dependent but is still translationally invariant. The systems stay homogeneous and only pair quantities like $g(r, t)$ carry the dynamics (we assume there is no spontaneous breaking of translation invariance).

We derived the Euler-Lagrange equations of motion for $g(r, t)$ for the simplest case, where elementary diagrams in the HNC relations are omitted, and only pair correlations $u_2$ are taken into account in the variational ansatz for the many-body wave function (Jastrow-Feenberg ansatz). This simple version is long known in various formulations for ground state calculations [58, 59] and is sufficient for not too strongly correlated Bose systems (but not sufficient for quantitative predictions of energy and structure of e.g. liquid $^4$He).

The dynamics of the pair distribution $g(r, t)$ of a Rydberg-dressed 3D gas is similar to results of previous tVMC studies of bosons on a deep 2D lattice, described by the Bose-Hubbard model [31], and bosons in 1D [32]. After a weak quench a "wave" travels to larger distances.

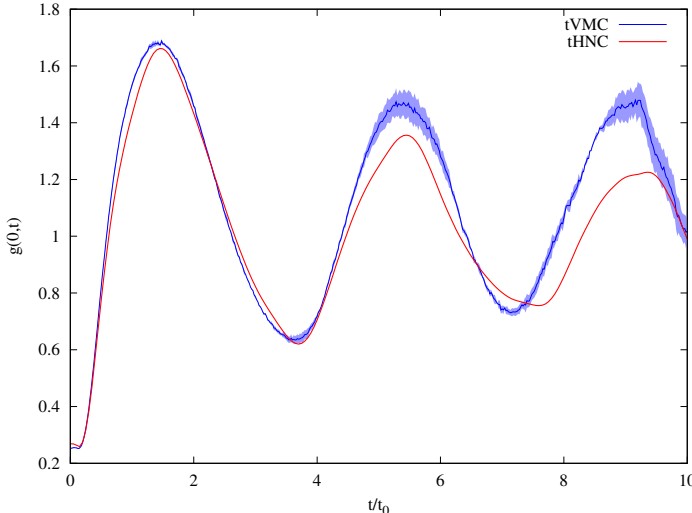

Figure 7: For the quench of the 2D Bose gas in Fig. 6, we compare the roton-induced oscillation of $g(r, t)$ for $r = 0$ between tHNC (red) and tVMC (blue). The shaded area shows the stochastic error of the tVMC result.

Unlike for the dynamics on a lattice, we observe no light-cone restriction for the speed of propagation since our group velocity is not bounded from above. Ripples of very high wave number indeed move very fast towards larger distances. We stress that these waves are not density fluctuations (which would break translation symmetry) but fluctuations of the *pair* density.

There is a qualitative change in the behavior of $g(r, t)$ for a strong quench. If the interaction after the quench is strong enough that a linear response calculation predicts a roton excitation, i.e. a non-monotonous dispersion relation, $g(r, t)$ exhibits long-lasting oscillations for small $r$ that do not decay within our calculation time windows. This can be readily understood as the creation of a pair of rotons with opposite momentum (total momentum is conserved by an interaction quench). The group velocity of rotons vanishes, therefore the oscillations of $g(r, t)$ due to rotons do not propagate to larger $r$; furthermore the density of states for rotons diverges, therefore exciting roton pairs is very efficient. The argument is equally valid for the maxon, i.e. the local maximum of the dispersion relation. Finally, since the quench creates a pair of rotons or maxons (a roton-maxon pair would violate momentum conservation), the oscillations have twice the roton or maxon frequencies. Indeed, for these strong quenches the power spectra have peaks at almost these frequencies, and not much strength at other frequencies. The peaks are slightly off from twice the roton or maxon frequency due to nonlinear effects, which we confirmed by smaller, hence more linear-response-like, jumps in the interaction quench.

Apart from the Jastrow-Feenberg ansatz with pair correlations, the tHNC equations contain two approximations, the already mentioned omission of elementary diagrams, and an approximation for the three-body distribution function $g_3(\mathbf{r}_1, \mathbf{r}_2, \mathbf{r}_3)$, where we chose the straightforward Kirkwood superposition approximation. In order to assess these two approximations, we performed tVMC simulations of interaction quenches. We restricted ourselves to one and two dimensions due to the high computational cost of tVMC. Overall we see good agreement between tHNC and tVMC. In 1D we compared for two different quenches, finding excellent agreement for the weaker quench. As the interaction strength after the quench increases the agreement worsens somewhat, but all the main features such as the roton-induced oscillations in the 2D comparison are captured already by tHNC.

Comparison with tVMC or other time-dependent many-body methods may be prohibitive, such as for the study of the long-range behavior of $g(r, t)$ in 3D. In such a case, the errors in tHNC can only be studied and reduced by improving tHNC itself. For strong interactions, we should (i) incorporate elementary diagrams – this can be done approximately in the same way as in ground state HNC-EL calculations; (ii) improve upon the Kirkwood superposition approximation (e.g. use the systematic Abe expansion [41]) or compare with other approximations like the convolution approximation [34]; (iii) include triplet correlations $u_3(\mathbf{r}_1, \mathbf{r}_2, \mathbf{r}_3)$ in the variational ansatz (also for tVMC), as has been done for ground state and linear response calculations [29]. The latter improvement of tHNC will require completely new approaches how to solve the resulting high-dimensional equations of motion.

Like the ground state HNC-EL method, tHNC can be generalized to inhomogeneous and/or anisotropic systems. The latter is important for dipolar Bose gases, while most quantum gas experiments are in harmonic or optical traps rather than in box traps, and thus require an inhomogeneous description; even in case of an interaction quench of a homogeneous Bose gas, the quench may trigger spontaneous breaking of translational invariance. The diagrammatic summations used in the ground state HNC-EL method have been generalized to off-diagonal properties like the one-body density matrix to study off-diagonal long-range order, i.e. the Bose-Einstein condensed fraction [60]. We will generalize this to the time-dependent case. Further down the road, we plan to include three-body correlations $u_3$ at least approximately, known to improve ground states of highly correlated systems [26]. This opens a new scattering channel, where an interaction quench can generate excitation triplets, not just pairs, with total momentum zero. In the case of quenches to rotons, the oscillation pattern of $g(r, t)$ will be more complex because its power spectrum can contain frequencies corresponding to the energies of three rotons. We are also generalizing tHNC to anisotropic interaction in order to study the non-equilibrium dynamics of dipolar quantum gases. Studying the long-time dynamics after a nonlinear perturbation, such as a strong interaction quench, requires numerical stability for very long times, which is challenging for nonlinear problems like solving the tHNC equations or the equations of motion of tVMC. A small time step is required to achieve long-time stability for both tHNC and tVMC. However, tHNC is computationally very cheap and calculations of the long-time dynamics are feasible. In order to delay artifacts from reflections at domain boundaries, either the spatial domain must be chosen very large, or absorbing boundary conditions are implemented.

## Acknowledgments

We acknowledge discussions with Eckhard Krotscheck, Markus Holzmann, and Lorenzo Cevolani. The tVMC simulations were supported by the computational resources of the Scientific Computing Administration at Johannes Kepler University.

## A  Derivation of the tHNC equation

We derive the tHNC equation of motion (11) using the time-dependent variational principle $\delta \mathcal{S} = 0$, where $\mathcal{S}$ is the action defined in eq.(4).

The action is given by eqns. (4), (5), and (6) as

$$\mathcal{S} = \mathcal{S}_g + \mathcal{S}_2 + \mathcal{S}_3 \,,$$

with

$$S_g = \frac{\rho}{2} \int\limits_{t_0}^{t} dt' \int d^d r \left[ v(r,t') - \frac{\hbar^2}{4m} \Delta u(r,t') \right] g(r,t'),$$

$$S_2 = \frac{\hbar}{2} \int\limits_{t_0}^{t} dt' \int d^d r \, \dot\varphi(r,t') g(r,t') + \frac{\hbar^2}{2m} \rho \int\limits_{t_0}^{t} dt' \int d^d r \, \mathbf{v}(\mathbf{r},t')^2 g(r,t'),$$

$$S_3 = \frac{\hbar^2}{2m} \rho^2 \int\limits_{t_0}^{t} dt' \int d^d r \, d^d r' \, \mathbf{v}(\mathbf{r},t') \cdot \mathbf{v}(\mathbf{r}',t') g_3(r,r',|\mathbf{r}-\mathbf{r}'|,t'),$$

where $\mathbf{v}(\mathbf{r},t') \equiv \nabla\varphi(r,t')$.

The Euler-Lagrange equations (9) and (10) require the functional differentiation of $S_g$, $S_2$, and $S_3$ with respect to $g(r,t)$ and $\varphi(r,t)$. Apart from time integration, $S_g$ is the same expression as the energy expectation value of the ground state HNC-EL method. It does not depend on $\varphi(r,t)$ and the variation with respect to $g(r,t)$, using the HNC relation (7), can be found in reviews on HNC-EL, e.g. Ref. [28]. The derivatives of $S_2$ are

$$\frac{\delta S_2}{\delta g(r,t)} = \frac{\hbar}{2} \dot\varphi(r,t) + \frac{\hbar^2}{2m} \mathbf{v}(\mathbf{r},t)^2,$$

$$\frac{\delta S_2}{\delta \varphi(r,t)} = -\frac{\hbar}{2} \dot g(r,t) - \frac{\hbar^2}{2m} \nabla \left[ g(r,t) \cdot \mathbf{v}(\mathbf{r},t) \right].$$

In $S_3$ we employ the Kirkwood superposition approximation (8)

$$S_3 = \frac{\hbar^2}{2m} \rho^2 \int\limits_{t_0}^{t} dt' \int d^d r \, d^d r' \, \mathbf{f}(\mathbf{r},t') \cdot \mathbf{f}(\mathbf{r}',t') g(|\mathbf{r}-\mathbf{r}'|,t'),$$

with $\mathbf{f}(\mathbf{r},t') \equiv g(r,t') \mathbf{v}(\mathbf{r},t')$. The derivatives are

$$\frac{\delta S_3}{\delta g(r,t)} = \frac{\hbar^2}{m} \mathbf{v}(\mathbf{r},t) \cdot \nabla\gamma(r,t) + \frac{1}{2}\beta(r,t),$$

$$\frac{\delta S_3}{\delta \varphi(r,t)} = -\frac{\hbar^2}{m} \nabla \left[ g(r,t) \cdot \nabla\gamma(r,t) \right].$$

where $\beta(r,t)$ and $\nabla\gamma(r,t)$ are defined in eqns. (12) and (13), respectively.

Putting everything together, equation (9) becomes, after multiplying by 2,

$$0 = -\frac{1}{\sqrt{g(r,t)}} \frac{\hbar^2}{m} \nabla^2 \sqrt{g(r,t)} + v(r,t) + w_I(r,t) + \hbar\dot\varphi(r,t) + \frac{\hbar^2}{m} [\nabla\varphi(r,t)]^2$$

$$+ 2\frac{\hbar^2}{m} \nabla\varphi(r,t) \cdot \nabla\gamma(r,t) + \beta(r,t). \tag{A.1}$$

If we kept only the first line of the equation, we would recover the ground state HNC-EL equation (where $\varphi = 0$, of course). The induced potential $w_I(r,t)$ describing phonon-mediated interactions can be found in Ref. [28]. Equation (10) becomes

$$0 = -\frac{1}{2}\hbar\dot g(r,t) - \frac{\hbar^2}{m} \nabla \left[ g(r,t) \cdot \nabla(\varphi(r,t) + \gamma(r,t)) \right]. \tag{A.2}$$

When we multiply eq. (A.1) with $\sqrt{g(r,t)}$ and eq. (A.2) with $i/\sqrt{g(r,t)}$, add the two equations, and multiply the resulting equation with $e^{i\varphi(r,t)}$, we obtain the final form (11) of the tHNC equation for the effective pair wave function $\psi(r,t) = \sqrt{g(r,t)} e^{i\varphi(r,t)}$ that we solve numerically.

# B The tVMC method

In tVMC [30–32], we use the same Jastrow-Feenberg ansatz of equation (3) as in tHNC, and describe the time dependence of the wavefunction via a set of $P$ complex variational parameters $\alpha(t) = \{\alpha_1(t), \alpha_2(t), \ldots, \alpha_P(t)\}$, which are coupled to local operators $\mathcal{O}_m(\mathbf{r}_1, \ldots, \mathbf{r}_N)$. In our implementation (more details can be found in [57]), these local operators are real and represented by third order B-splines $B_m(r)$ centered on a uniform grid in the interval $[0, L/2]$, and are given by $\mathcal{O}_m(\mathbf{r}_1, \ldots, \mathbf{r}_N) = \sum_{k<l} B_m(r_{kl})$. The real and imaginary part of the pair correlation function $u_2(r)$ can then be written as

$$u(r) = \sum_m^P B_m(r)\alpha_m^R(t), \qquad \varphi(r) = \sum_m^P B_m(r)\alpha_m^I(t),$$

where $\alpha_m^R$ and $\alpha_m^I$ are the real and imaginary part of $\alpha_m$. The time evolution of the variational wavefunction is obtained by solving the coupled system of equations (see [30])

$$i\sum_n S_{mn}\dot{\alpha}_n = \langle \mathcal{E}\mathcal{O}_m \rangle - \langle \mathcal{E} \rangle \langle \mathcal{O}_m \rangle, \tag{B.1}$$

with the correlation matrix $S_{mn} = \langle \mathcal{O}_m\mathcal{O}_n \rangle - \langle \mathcal{O}_m \rangle \langle \mathcal{O}_n \rangle$ and the local energy $\mathcal{E} = \frac{H|\Phi\rangle}{|\Phi\rangle}$. The expectation values $\langle \ldots \rangle$ are estimated using Monte Carlo integration by sampling from the trial wavefunction $\Phi(\mathbf{r}_1, \ldots, \mathbf{r}_N, t) = \exp\left(\sum_{k<l} u_2(r_{kl}, t)\right)$. Because of the translational invariance of the studied system we do not need to take into account a one-body part $u_1(r, t)$ in the wavefunction, unlike in Ref. [57] where the dynamics of a Bose gas in a 1D optical lattice has been simulated.

**Simulation parameters in 1D:** In tVMC, we approximate the Jastrow pair-correlation function $u(r)$ using a cubic spline function with $P = 400$ complex weights, corresponding to the time-dependent variational parameters $\alpha_m(t)$, as given above. The time propagation is performed with a time step of $\Delta t = 2 \cdot 10^{-4}\, t_0$, and $N_{MC} = 1250$ uncorrelated samples are used for calculating the expectation values required for time propagation. The results are converged with respect to the spatial and temporal numerical resolution as we see no changes in the simulation results upon increasing $P$ or decreasing $\Delta t$.

Performing tVMC simulations comes of course with a large computational workload compared to tHNC calculations. While about 560 CPU hours are required to run the discussed tVMC simulations in 1D, tHNC takes about 5 CPU minutes for evolving the pair distribution function for the same time window $t \in [0, 20]t_0$ and for an even larger $r$-domain, which reduces unphysical boundary reflections. However, as most Monte Carlo methods, tVMC can be highly parallelized, which makes it feasible in terms of real computing time.

**Simulation parameters in 2D:** In the 2D tVMC simulations we use $N = 900$ particles, corresponding to a simulation box size $L = 106.347\, r_0$. We use $P = 300$ variational parameters, a time step of $\Delta t = 2 \cdot 10^{-4}\, t_0$ and $N_{MC} = 1250$ uncorrelated samples for estimating the expectation values needed to solve equation (B.1).

For the 2D tVMC simulations with $N = 900$ particles the computational time amounts to 14.000 CPU hours, stressing one more time the benefits of tHNC simulations concerning the computational workload, which only amounts to approx. 30 CPU hours.

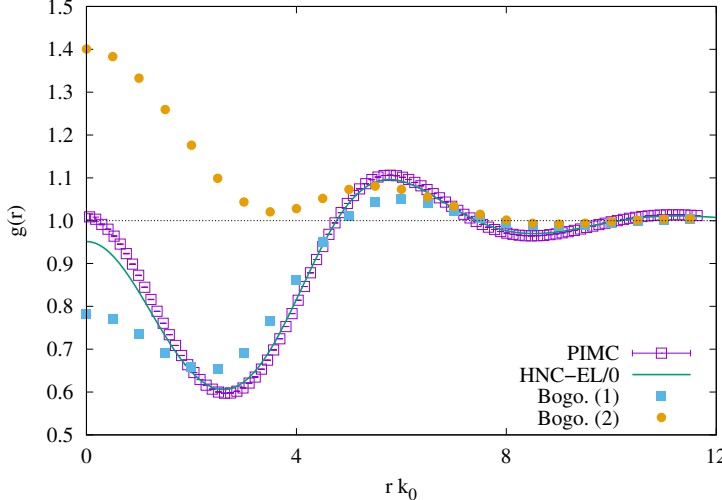

Figure 8: Comparison of the ground state pair distribution function $g(r)$ obtained from HNC-EL (line), from exact PIMC in the limit of low temperature (open squares), from the Bogoliubov method (filled circles), and an approximate Bogoliubov method (filled squares). See text for details.

## C  Comparison with Bogoliubov approximation

Interaction quenches in Bose gases have been studied with the Bogoliubov method which accounts for correlations as a perturbation ("quantum fluctuations") of the mean field approximation. The pair distribution function $g(r)$ in Bogoliubov approximation [61] has been generalized to dynamical problems in Refs. [48, 62]. Particularly for quenches in Rydberg-dressed Bose gases this has been used, with a further approximation, in Ref. [47].

In Fig. 8 we compare the result for the ground state $g(r)$ for the 3D Rydberg-dressed Bose gas for $R = 4$ and $U = 3$ obtained with four different methods: exact path integral Monte Carlo (PIMC) in the limit of low temperature, such that the Bose gas is effectively in the ground state [46]; the HNC-EL/0 method which is the ground state limit of the time-dependent tHNC method used in this paper; and two variants of the Bogoliubov method: first, the full expression for $g(r)$ used in Refs. [48, 62] for dynamics, referred to as Bogo.(1) in Fig. 8; and secondly, an approximate expression used in Ref. [47] for dynamics, referred to as Bogo.(2) in Fig. 8. From Fig. 8 we see that HNC-EL/0 result for $g(r)$ (line) reproduces the exact PIMC result (open squares) very well, apart from small deviations for $r \to 0$. On the other hand, both Bogoliubov results deviate quite significantly from the exact PIMC result. The comparison demonstrates that correlations must be incorporated non-perturbatively in this case, while for weakly interacting Bose systems the Bogoliubov approximation may be sufficient.

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
