# Peer review of "Interaction quenches in Bose gases studied with a time-dependent hypernetted-chain Euler-Lagrange method"

_SciPost Physics, doi:SciPost Phys. 18, 123 (2025)_

## Round 1 · Referee Report · Anonymous (Referee 1) · 2023-2-1

Strengths

1 - The problem of time dynamics in quantum many-body systems is challenging as there are no general well-developed tools for addressing it while experimentally such dynamics can be studied. In that context, a time-dependent version of HNC method provides a useful numerical method of addressing time-dependent problems in a non-perturbative way
2 - The tHNC method is verified in 1D and 2D geometry in comparison to tVMC predictions. There is a good agreement for the first few oscillations after the quench.

Weaknesses

1 The structure of the article can be improved. It would be more natural to present first the methodology, then the verification (1D and 2D), and leave the new results (3D) for the last.

2 There are some unnecessary technical details given while the physical motivation is missing.

Report

Authors study the problem of finding the time dependence in a quantum many-body system which is on its own very interesting and difficult task, as there are no standard and well-established techniques for doing so numerically. A method based on hypernetted-chain approach is developed and applied to the problem of interaction quenches in one, two, and three dimensions. While the developed method is interesting I believe presentation can be significantly improved.

Requested changes

Main comments:

1 The way the article is structured now, first 3D results are shown and later verification of the method is done in 1D and 2D. It feels more natural to show first verification of the method and later apply it to some system where no alternative results are present. 2 Is it possible to predict the error of the obtained results within the used theory? It might be anticipated that the error is larger for larger times. Is it possible to see that? 3 In the benchmark tests, tHNC results are compared to predictions of tVMC method (for example Fig. 8). For small times there is a good agreement, for larger times differences are visible. Is it possible to add error bars to both theories to make the comparison quantitative? 4 Comment why the imaginary part of $u_2(r)=u(r)+2i\phi(r)$ has a factor of two as compared to the real part 5 When energy is given in units of $E_0$, it is not clear if energy per particle or defect energy is assumed. 6 Below Fig. 1, pair distribution is given for different values of $R/r_0$. It might be useful as well to report the smallest and the largest corresponding density, i.e. $\rho r_0^3$. 7 Fig. 2, it is commented that no light cone boundary is observed. Still, the actual boundary looks pretty linear for practical purposes. It might be useful to draw the light cone defined by the speed of sound in the bulk with a dashed line for comparison. 8 Fig. 2, color coding (blue-red) is not explained. 9 "The effect of non-linearity is demonstrated ...". Please provide a more detailed explanation of what is meant by a linear / nonlinear effect. 10 Sections on 1D and 2D comparison provide technical details which can be moved to the appendix. Instead, it would be good to state the physical motivation which systems are considered and why.

Minor comments

11 Visually it does not look nice to start a phrase with a lowercase letter ("tHNC ...") or symbols ("${\cal L}$...", "$g(r,t)$", "$g$", "g_3", "$w_I$", "$\beta$", "$\gamma$", etc) . Please rephrase. 12 "We do not include an external potential because we only consider homogeneous systems" Explain better the reasoning as right now it is tautologic. 13 "In the case that the perturbation is a change of perturbation, the only difference .. is the time-dependence of perturbation". Tautology once again. 14 "Note that $\gamma$ in eq.(10) is to be calculated from its gradient" Rephrase 15 "When two particles are closer than ... they are only weakly repelled because $u$ becomes flat", explain the physical reason (Rydberg blockade?) rather than the mathematical property of the used model 16 "For the 1D comparison we use a square well potential", just saying "potential" might be ambiguous as applied to external potential or interaction potential. 17 "We approximate the Jastrow pair-correlation ... with 400 splines". Is it a single spline function containing 400 parameters or are there 400 spline functions? 18 For some reason Fermi units are used in 2D and 3D but not in 1D, although this is a matter of choice and can be left as it is.

  • validity: good
  • significance: high
  • originality: good
  • clarity: good
  • formatting: good
  • grammar: excellent

Author:  Robert Zillich  on 2025-01-24  [id 5147]

(in reply to Report 1 on 2023-02-01)
Category:
answer to question

Dear Editors,

we thank the referees for their helpful reports, and apologize for the lengthy “awaiting resubmission” status of our
manuscript on the SciPost submission web site. We have finally revised the manuscript heeding the advice of the two
referees.

We attach the reply to their suggestions, questions, and criticism as pdf files.
The referee reports are italicized and our replies are non-italicized.

Attachment:

reply1.pdf

---

## Round 1 · Referee Report · Anonymous (Referee 2) · 2023-4-18

Strengths

  • the introduced tHNC method has computational advantage over existing approaches
  • the approximations of tHNC are clearly stated and ways for further improvement of the method are outlined
  • tHNC is used to tackle interesting and hard non-equilibrium many-body problems

Weaknesses

  • the range of the validity of the approach is not fully clear
  • some minor aspects of the presentation could be improved

Report

The authors develop a tHNC method for non-equilibrium many-body problems and employ it in investigations of quench dynamics of Bose gases. The method is presented in a clear and well-justified manner, with emphasis on the underlying assumptions. Application of the tHNC method to 3D gas of Rydberg-dressed atoms yields insights about interaction quenches in the system, and the authors propose plausible explanations for the observed phenomena. Comparison of tHNC method with tVMC for 1D and 2D systems allows to obtain certain intuitions about the range of validity of the introduced approach. The novel method proposed in the present manuscript allows for non-perturbative insights into dynamics of "not too strongly correlated" many-body systems. This hard problem is both conceptually interesting and of practical importance. Therefore, I recommend that the manuscript "Interaction quenches in Bose gases studied with a time-dependent hypernetted-chain Euler-Lagrange method" be published in SciPost Physics once the authors address the following minor remarks.

  1. The tHNC method does not require Monte Carlo sampling and hence it is significantly faster than the tVMC. It would be helpful if authors included some estimates of computational resources needed to perform the simulations from Sec. III, both for tHNC and tVMC.

  2. Data shown in Fig. 6 and 7 indicate that tHMC gets less reliable for larger interaction amplitudes, especially for longer times. How can we be sure that data presented in Fig. 2 for the larger values of $R/r_0$ and times up to $t/t_0=20$ (or for $t/t_0=40$ in Fig. 3) are in any correspondence with the actual dynamics of the system?

  3. Page 6. the meaning of "nonlinear process" is not clear but is important to understand the reasoning of the authors.

  4. Can the authors provide an explanation for the suppression of the excitation of two maxons during the $4r_0 \to 4.5r_0$ quench, perhaps due to energetics of the system?

  5. The authors mention mean field studies of quench dynamics in Rydberg dressed Bose gases [34-36]. For the sake of self-completeness of the manuscript, it would be useful to contain a more in-depth comparison of the results of tHNC with the earlier simpler approaches.

  6. In the introduction section, the addition of references to TDVP (alongside of TEBD) and some standard works on BEC, quench experiments, Feshbach resonances would help make the description more concrete and provide a better context for the study.

Requested changes

To address the aforementioned questions 1-6.

  • validity: high
  • significance: high
  • originality: high
  • clarity: high
  • formatting: good
  • grammar: excellent

Author:  Robert Zillich  on 2025-01-24  [id 5148]

(in reply to Report 2 on 2023-04-18)

Dear Editors,

we thank the referees for their helpful reports, and apologize for the lengthy “awaiting resubmission” status of our
manuscript on the SciPost submission web site. We have finally revised the manuscript heeding the advice of the two
referees.

We attach the reply to their suggestions, questions, and criticism as pdf files.
The referee reports are in blue and our replies are in black.

Attachment:

reply2.pdf

---

## Round 2 · Referee Report · Anonymous (Referee 1) · 2025-1-28

Report

The Authors have appropriately addressed all raised comments. Also the text was reorganized and extended. I find that this improved the quality of the presentation and now the Manuscript meets the high requirements of the journal.

I recommend the publication of the Manuscript.

The only minor comment I have is that wording "average neighbor distance" is not commonly used as compared to "mean interparticle distance". Also it might something else, for example, in a perfect hexanal 2D crystal, each atoms has 6 neighbors at the same distance (lattice spacing), while distance, related to the density is different.

Recommendation

Publish (easily meets expectations and criteria for this Journal; among top 50%)

---

## Round 2 · Referee Report · Anonymous (Referee 2) · 2025-2-14

Report

The authors have convincingly replied to all my comments. I recommend the publication of the manuscript "Interaction quenches in Bose gases studied with a time-dependent hypernetted-chain Euler-Lagrange method" in SciPost Physics.

Recommendation

Publish (easily meets expectations and criteria for this Journal; among top 50%)

---

## Round 2 · Author Response

Dear Editors,

we thank the referees for their helpful reports, and apologize for the lengthy “awaiting resubmission” status of our
manuscript on the SciPost submission web site. We have finally revised the manuscript heeding the advice of the two
referees.

We attach the reply to their suggestions, questions, and criticism as pdf files uploaded via
the "Reply to the above Report" link.
The referee reports are in blue and our replies are in black.

---

## Round 2 · List of Changes

We added a comparison of the computational effort of tHNC and tVMC.
We explain what we mean by ``nonlinear process''
We provide a comparison between the Bogoliubov approximation, the HNC approximation and the exact result for $g(r)$ in the ground state in a new appendix C, including a new Fig.8.
We improved Fig.2 and its caption: adding a color bar, adding the sound cone and explaining the color scheme;
At the start of the results sections we elaborated a bit more on how the result section is structured
At the start of section 3.1 we give a physical picture of the Rydberg-dressed interaction.
We moved the technical details of the tVMC simulations to the appendix.
We extended the discussion of the ``light cone'' ideas considerably, describing and explaining the additional light cone lines in Fig.1.
We combined former Fig.1 and 4 into one Fig. with two panels, because in the new discussion of the light cone we need to refer to Fig.4 earlier in the text than in the old manuscript.
We updated Fig.7 (former Fig.8) by showing the stochastic error.
We converted the inline equation for the action S into equation (4) in order to reference this definition in app.A
In app. A, we explicitly show the time dependence of all quantities instead of omitting it
We added a reference for Feshbach resonances in the introduction
various small changes in formulation

New References:
A. Kerman and S. Koonin, Annals of Physics 100, 332–358 (1976).
Haegeman et al, PRL 107, 070601 (2011)
G. Pupillo et al, PRL 104, 223002 (2010)
C. Chin et al, Rev. Mod. Phys. 82, 1225 (2010)
L. Madeira and V. S. Bagnato, Symm. 14, 678 (2022)
F. Dalfovo, S. Giorgini, L. P. Pitaevskii and S. Stringari, Rev. Mod. Phys. 71, 463 (1999)
O. Morsch and M. Oberthaler, Rev. Mod. Phys. 78, 179 (2006).
A. Polkovnikov et al., Rev. Mod. Phys. 83, 863 (2011)
T. Langen, R. Geiger and J. Schmiedmayer, Annu. Rev. Condens. Matter Phys. 6, 201 (2015)
M. Cheneau et al., Nature 481, 484 (2012)
P. Makotyn, Nat. Phys. 10, 116(2014)
N. Navon et al.,Nat. Phys. 17(12), 1334 (2021)

---

## Editorial Decision

published